# MiCas9 increases large size gene knock-in rates and reduces undesirable on-target and off-target indel edits

Linyuan Ma[1], Jinxue Ruan[1], Jun Song[1], Luan Wen[1], Dongshan Yang[1], Jiangyang Zhao[2], Xiaofeng Xia[2], Y. Eugene Chen[1✉], Jifeng Zhang[1✉] & Jie Xu[1✉]

Gene editing nuclease represented by Cas9 efficiently generates DNA double strand breaks at the target locus, followed by repair through either the error-prone non-homologous end joining or the homology directed repair pathways. To improve Cas9's homology directed repair capacity, here we report the development of miCas9 by fusing a minimal motif consisting of thirty-six amino acids to spCas9. MiCas9 binds RAD51 through this fusion motif and enriches RAD51 at the target locus. In comparison to spCas9, miCas9 enhances double-stranded DNA mediated large size gene knock-in rates, systematically reduces off-target insertion and deletion events, maintains or increases single-stranded oligodeoxynucleotides mediated precise gene editing rates, and effectively reduces on-target insertion and deletion rates in knock-in applications. Furthermore, we demonstrate that this fusion motif can work as a "plug and play" module, compatible and synergistic with other Cas9 variants. MiCas9 and the minimal fusion motif may find broad applications in gene editing research and therapeutics.

[1] Center for Advanced Models for Translational Sciences and Therapeutics, University of Michigan Medical School, 2800 Plymouth Road, Ann Arbor, MI 48109, USA. [2] Research & Development, ATGC Inc., 100 E Lancaster Avenue, LIMR Building Lab129, Wynnewood, PA 19096, USA. ✉email: echenum@umich.edu; jifengz@umich.edu; jiex@med.umich.edu

Gene editing nucleases (GENs) represented by CRISPR (clustered regularly interspaced short palindromic repeats)/Cas9 (CRISPR-associated protein 9) and others have become mainstream tools in biomedical research and offer promising therapeutic potentials[1,2]. These GENs efficiently create double-strand breaks (DSBs) at the target locus in the genome. The majority of the DSBs are repaired by the non-homologous end joining (NHEJ) mechanisms leading to insertions and deletions (indels)[3]. Homology directed repair (HDR), which generally occurs at a significantly lower frequency than NHEJ[4,5], is exploited to achieve knock-in (KI).

The use of GENs such as *Streptococcus pyogenes* Cas9 (spCas9) along with single-stranded oligodeoxynucleotide (ssODN) templates has allowed high precise gene editing (ss-PGE) rates often at two-digit efficiencies, through the single-strand annealing (SSA) and/or the synthesis-dependent strand annealing (SDSA) pathways[6–10]. The use of GENs also improved large size double-stranded DNA (ds-DNA) mediated KI (ds-KI) through the homologous recombination (HR) pathway, albeit the ds-KI rate is still low and needs to be improved[11–13]. In both ss-PGE and ds-KI applications, undesirable on-target and off-target indels pose a potential concern for therapeutic applications, which also needs to be addressed.

There are two main engineering strategies to improve spCas9. The first is to fuse key modules implicated in the HDR pathway to spCas9 (HDR-fusion Cas9s). Reported fusion motifs include Geminin[14], CtIP[15], mSA[16], Rad52 (ref. [17]), DN1S[18], among others[19–21]. These HDR-fusion Cas9 variants generally increase ds-KI rates, but none has been reported to simultaneously reduce on-target and off-target indel rates without compromising ss-PGE rates. In addition, the fusion partners are often of large sizes, most consisting of more than 100 amino acids (AA), a disadvantageous feature that may affect packaging and transduction efficiencies in vitro and in vivo via AAV. The second major Cas9 improving strategy is achieved through point mutation to improve specificity (specificity-improving Cas9s) without changing the size of spCas9, which include eSpCas9 (ref. [22]), SpCas9-HF1 (ref. [23]), xCas9 (ref. [24]), HypaCas9 (ref. [25]), and HiFiCas9 (ref. [26]). While specificity-improving Cas9s are capable of reducing off-target indel rates, they have little effect on enhancing ds-KI rates or reducing on-target indel rates.

In the present work, we develop a small size HDR-fusion variant, miCas9. Through its fusion motif Brex27, miCas9 binds RAD51 and enriches RAD51 at the target locus. Comparing to spCas9, miCas9 increases ds-KI rates, reduces off-target indel rates, and reduces undesirable on-target indel events in ds-KI and ss-PGE applications.

## Results

**Construction of miCas9**. One of the key events in the HR process is the deposition of RAD51 by BRCA2 to the resected single-stranded DNA (ssDNA) overhang at DSB to form RAD51/ssDNA nucleoprotein filaments, the ultimate molecular species that performs homology search and strand exchange underlying homologous recombination[27–29]. Prior studies have elucidated that a motif consisting of 36 AA encoded by BRCA2 Exon 27 (Brex27) binds RAD51, and stabilizes the RAD51/ssDNA nucleoprotein filaments[30]. Hence, we hypothesize that the fusion of Brex27 to spCas9 will lead to local enrichment of RAD51 as well as stabilization of the RAD51/ssDNA nucleoprotein filaments at the Cas9-induced on-target and off-target DSBs, which consequently may have beneficial effects on reducing undesirable on-target and off-target indel events and improving ds-KI rates (Fig. 1a). Based on this rationale, we constructed miCas9-expressing plasmid DNA (pDNA) and spCas9-expressing pDNA

on the same backbone using the same codon optimizations (Fig. 1b, c, and Supplementary Fig. 1a, b). In order to use Cas9s in the ribonucleoprotein (RNP) form, we expressed and purified Cas9 proteins in *E. coli* (Supplementary Fig. 1c). SpCas9 and miCas9 RNPs possesses similar DSB generation capacity, evidenced by in vitro digestion assay results (Supplementary Fig. 1d).

**MiCas9 increases KI rates in ds-KI applications**. We first compared the efficiencies to knock in the green fluorescent protein (GFP) gene to a safe harbor locus AAVS1 in human cells between miCas9 and spCas9. The ds-DNA donor template (GFP-donor-1k-AAVS1, Supplementary Fig. 2a and Supplementary Table 1) consists of the promoterless GFP coding sequence (989 bp) flanked by left (804 bp) and right (837 bp) homology arms (HAs). The percentage of GFP-positive cells (%GFP+ cells) determined by flow cytometry analysis was used as an indicator of KI rates, as previously reported[15,31,32].

When Cas9s were delivered in their plasmid DNA (pDNA) form to fibroblast cells along with targeting guide RNA (sg-AAVS1, Supplementary Table 1) and GFP-donor-1k-AAVS1 donors, the %GFP+ cells was 2.5-fold higher in the miCas9 group than in the spCas9 group (2.93% vs. 1.19%) (Fig. 2a). Similar levels of improvement were confirmed in other cell types including airway epithelial cells (AECs, 0.72–1.49%) and induced pluripotent stem cells (iPSCs, 0.54–1.16%) (Fig. 2a). The donor integration at this locus appeared to be precise in GFP+ fibroblast cells as there were no detectable non-precise integrations at either the left junction or the right junction, in both spCas9 and miCas9 groups (Supplementary Fig. 2b, c). Increasing the KI fragment size from 989 bp (1K) to 2469 bp (2K) and 3339 bp (3K) (GFP-donor-2k-AAVS1, GFP-donor-3k-AAVS1, Supplementary Table 1) while keeping the same HAs slightly decreased the overall %GFP+ cells, but the rates were consistently 2–3-fold higher in the miCas9 group than in the spCas9 group (Supplementary Fig. 2d).

We further demonstrate that such improvement holds true when Cas9s were used in their RNP form. Using gRNAs and donor templates (GFP-donor-1k-AAVS1) targeting the AAVS1 locus, miCas9 RNPs led to significantly higher percentage of GFP+ cells than spCas9 RNPs did in both fibroblast (14.16% vs. 9.63%) and Jurkat cells (30.33% vs. 10.03%) (Fig. 2a).

To confirm that the ds-KI rate improvement achieved by miCas9 over spCas9 is not locus specific, we conducted a similar set of experiments at another locus ROSA26 (ref. [33]), using corresponding gRNA and ds-DNA donor template (Supplementary Table 1). Consistently, compared to spCas9, the use of miCas9 improved GFP KI rate by 2–3-folds in fibroblast cells (1.35–3.04%), AECs (0.55–1.32%) and iPSCs (0.60–1.53%) (Supplementary Fig. 3a) when used in the pDNA form. When used in the form of RNPs, miCas9 also led to higher GFP+ cells than spCas9 RNPs did in both fibroblast (16.02% vs. 12.91%) and Jurkat cells (15.00% vs. 7.24%) (Supplementary Fig. 3a).

These results show that miCas9 improves ds-KI rates at different loci in different human cell types by multiple folds.

**MiCas9 reduces on-target indel rates in ds-KI applications**. The design of miCas9 suggests that this modified GEN will not only improve HR-mediated ds-KI rates but will also reduce the on-target indel rates. We therefore proceeded to test this at multiple loci including AAVS1, ROSA26, ATF4, AHR, RAD21, and TGIF2. In addition to spCas9, two HDR-fusion Cas9s, Cas9-HE (HE) and Cas9-Geminin (GE) that have been shown to enhance ds-KI rates[14,15,21], were included in the comparison. Different

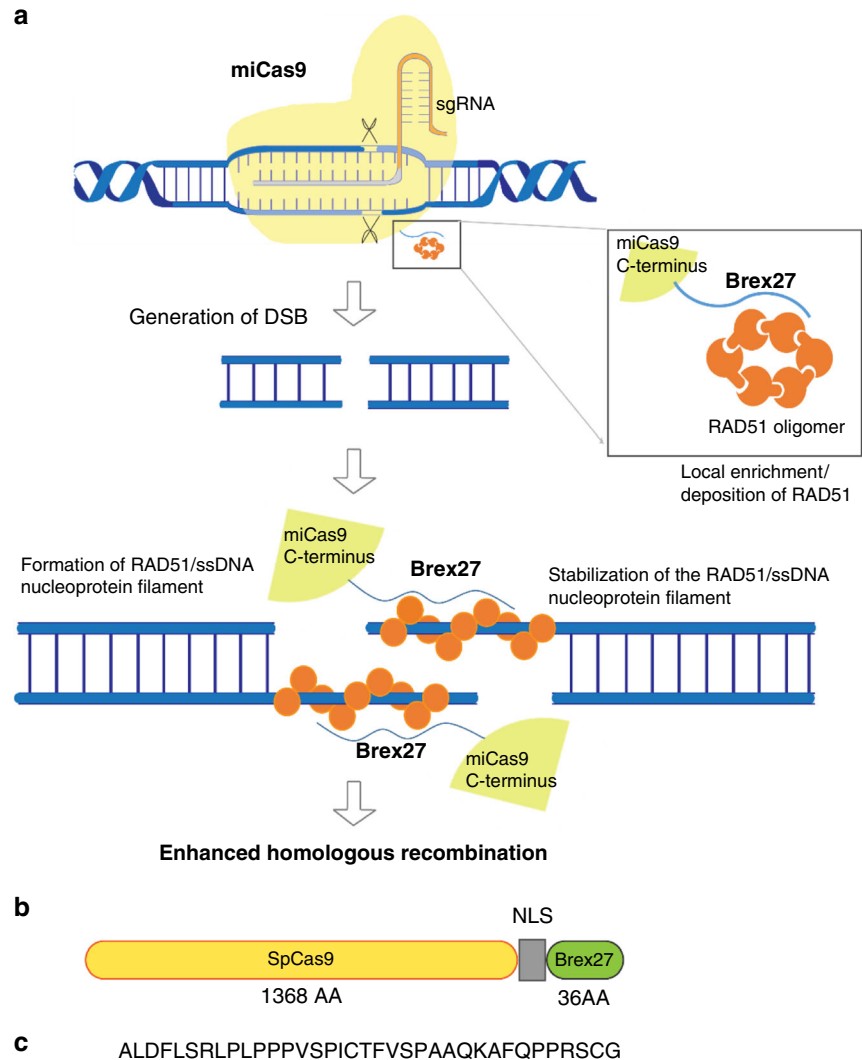

**Fig. 1 Illustration of miCas9 design. a** Rationale for miCas9 design. **b** Illustration of miCas9-expressing plasmid DNA construct. **c** Amino acid sequence of the Brex27 motif.

variants of Cas9 were delivered in their pDNA format along with gRNAs and GFP-expressing ds-DNA donors for each locus to fibroblast cells (Supplementary Table 1).

At the AAVS1 locus, comparable percentage of GFP⁺ cells were observed in the miCas9, Cas9-HE, and Cas9-GE groups (2.63–2.98%), all significantly higher than those in the spCas9 group (1.56%) (Fig. 2b). The on-target indel rate, revealed by next-generation sequencing (deep-seq) analysis, was lowest in the miCas9 (5.79%) group, followed by Cas9-HE (12.17%), and highest in the spCas9 (18.63%) and Cas9-GE (19.91%) groups (Fig. 2c).

Similar results were observed in the other five loci (ROSA26, ATF4, AHR, RAD21, and TGIF2). MiCas9, Cas9-HE, and Cas9-GE all resulted in an improved GFP KI rate than spCas9 did, but the lowest on-target indel rates were always achieved by miCas9, which was up to 75% lower than that by spCas9 and those by Cas9-HE and Cas9-GE (Supplementary Fig. 3b–f).

These results demonstrate that for ds-KI applications, miCas9 is superior to spCas9 in safety and efficacy performances by dramatically reducing the on-target indel rates and at the same time enhancing the KI rates. Comparing to two other HDR-fusion Cas9 variants (i.e. Cas9-HE and Cas9-GE), miCas9 possess the unique capacity to reduce on-target indel rates.

**MiCas9 reduces off-target indel rates.** Next we evaluated miCas9's capacity in reducing off-target indel rates. The AAVS1 and ROSA26 sgRNAs are not suitable for demonstrating this feature as their off-target rates by miCas9 or spCas9 are both low (Supplementary Fig. 4a, b). To better evaluate this, we examined several loci that are reportedly associated with substantial off-target rates by corresponding gRNAs, namely sg1-VEGFA and sg2-VEGFA targeting VEGFA, sg-FANCF2 targeting FANCF2, and sg-EMX1 targeting EMX1 (refs. [23,34]) (Supplementary Table 2) in Ad293 cells. We used miCas9 and spCas9 in their pDNA form because pDNA-mediated gene editing is known to be of higher off-target risks than those achieved by Cas9 RNP[6].

Deep-seq analysis reveals that compared to spCas9, miCas9 effectively reduced indel rates at all of the off-target loci examined (Fig. 3a, b and Supplementary 4c, c), including sg1-VEGFA-OT1 (34.70 to 3.30%), sg2-VEGFA-OT1 (3.9 to 0.34%), sg2-VEGFA-OT2 (13.4 to 1.9%), sg-EMX1-OT1 (46.7 to 31.1%), and sg-FANCF2-OT1 (25.1 to 4.3%). The on-target indels rates associated with these gRNAs are also lower in the miCas9 group than in the spCas9 group, similar to those observed in ds-KI application (Fig. 3a, b and Supplementary Fig. 4c, d). These results show that compared to spCas9, miCas9 dramatically reduced off-target indel events.

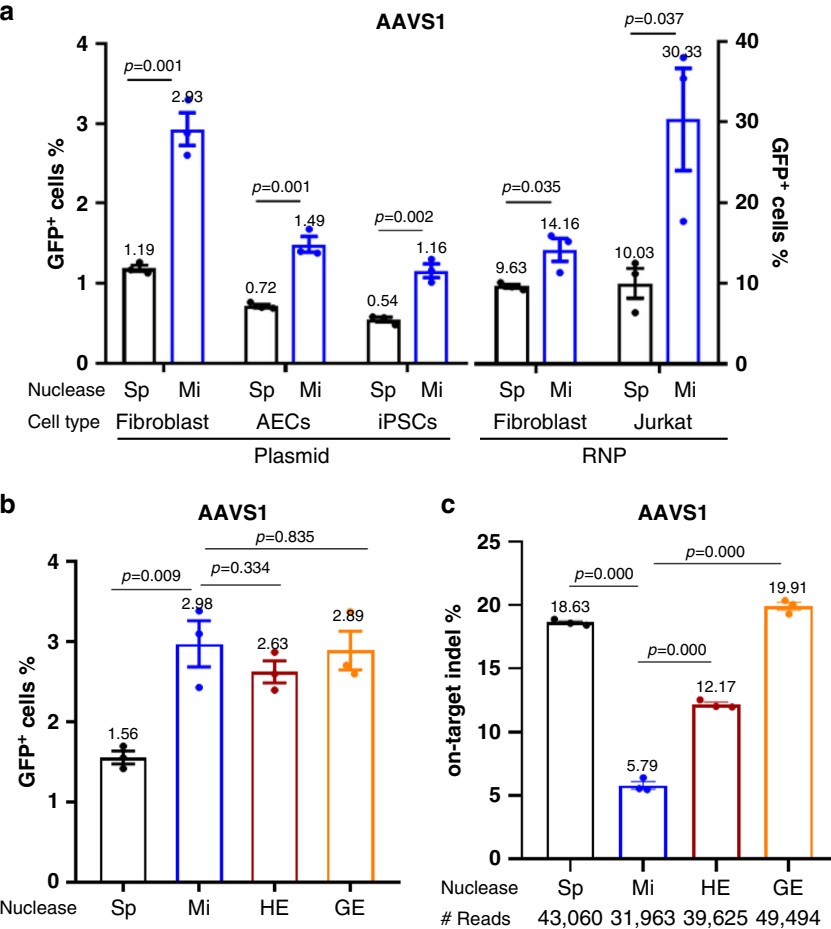

**Fig. 2 MiCas9 increases KI rates and reduces on-target indel rates in ds-KI applications. a** Percentage of GFP⁺ cells after GFP ds-KI to the AAVS1 locus in different cell types. **b** Percentage of GFP⁺ cells after GFP knock-in at the AAVS1 locus using different nucleases. **c** On-target indel rates at the AAVS1 locus after GFP ds-KI using different nucleases. AEC: airway epithelial cells, iPSCs: induced pluripotent stem cells, RNP: Cas9/gRNA ribonucleoprotein, Sp: spCas9, Mi: miCas9, HE: Cas9-HE, GE: Cas9-GE. #Reads: Average amplicon reads per sample. Three independent experiments were performed for each condition. Data are presented as mean ± standard error of means (SEM). Unpaired *t*-test (two tailed) was used to compare data using GraphPad Prism 8 software (GraphPad Software, Inc., San Diego, CA). Source data are available in the Source Data file.

To strengthen this finding, we next evaluated the off-target indel rates at Guide-seq[35] predicted loci for sg1-VEGFA and sg-FANCF2 in Ad293 cells (Fig. 3c, d). In addition to spCas9 and miCas9, we included Cas9-HE and Cas9-GE representing HDR-improving Cas9s, and HiFiCas9 and HypaCas9 representing specificity-improving Cas9s in the comparison.

For sg1-VEGFA, we evaluated nine Guide-seq predicted off-target (OT1 to OT9, Supplementary Table 2) loci (Fig. 3c). MiCas9 led to much lower indel rates in all these OT loci than spCas9 did. The extent of off-target indel reduction was similar between miCas9 and specificity-improving Cas9s (i.e. HypaCas9 and HiFiCas9). Cas9-HE, consisting of the CtIP fusion motif, also reduced off-target indel rates in general when compared to spCas9, but to a less extent compared to those by specificity-improving Cas9s and miCas9. Interestingly, Cas9-GE, consisting of a cell-cycle-modulating motif, did not show much effect, with indel rates similar to those observed in the spCas9 group.

For sg-FANCF2, we evaluated 10 Guide-seq predicted off-target (OT1–OT10, Supplementary Table 2) loci (Fig. 3d). Similar to the patterns revealed for sg1-VEGFA, miCas9 was as effective as HypaCas9 and HiFiCas9 in reducing off-target indel rates, whereas Cas9-HE had moderate effects and Cas9-Geminin had the least effect.

These results show that miCas9 effectively reduces undesirable off-target indel rates.

**MiCas9 reduces on-target indel rates without compromising efficacy in ss-PGE applications**. We then examined the effects of miCas9 on ss-PGE applications. We anticipated that miCas9 is capable of reducing undesirable on-target indel rates, as we have observed in ds-KI applications. We also wanted to learn if miCas9 has any effects on the PGE rates when ssODN donors are used.

MiCas9 and spCas9 were delivered in their RNP form to human iPSCs, along with corresponding gRNAs and ss-ODNs for generating point mutations at five different loci including EGFR, B2M, MYBPC3, NKX2.1, and SOD1 (Supplementary Table 3). Deep-seq was employed to analyze both PGE rates and on-target indel rates. The Noise/Signal (N/S) value was calculated as the indel/PGE rates at a given locus.

At all five loci, the PGE rates were similar between miCas9 groups and spCas9 groups (Fig. 4a and Supplementary Fig. 7a–d), eliminating the concern that miCas9 may adversely affect the efficacy of ss-PGE.

At four (i.e. EGFR, B2M, MYBPC3, and NKX2.1) out of these five (80%) loci, the on-target indel rates were significantly reduced by miCas9 compared to spCas9 (Fig. 4a and Supplementary Fig. 7a–d). For example, at the EGFR locus (Fig. 4a), the on-target indel rate was reduced from 21.4% in the spCas9 group to 2.96% in the miCas9 group, representing an 86% reduction [calculated as (high value–low value)/high value × 100]. At the B2M PGE locus (Fig. 4b), the on-target indel rate was reduced to below the detection

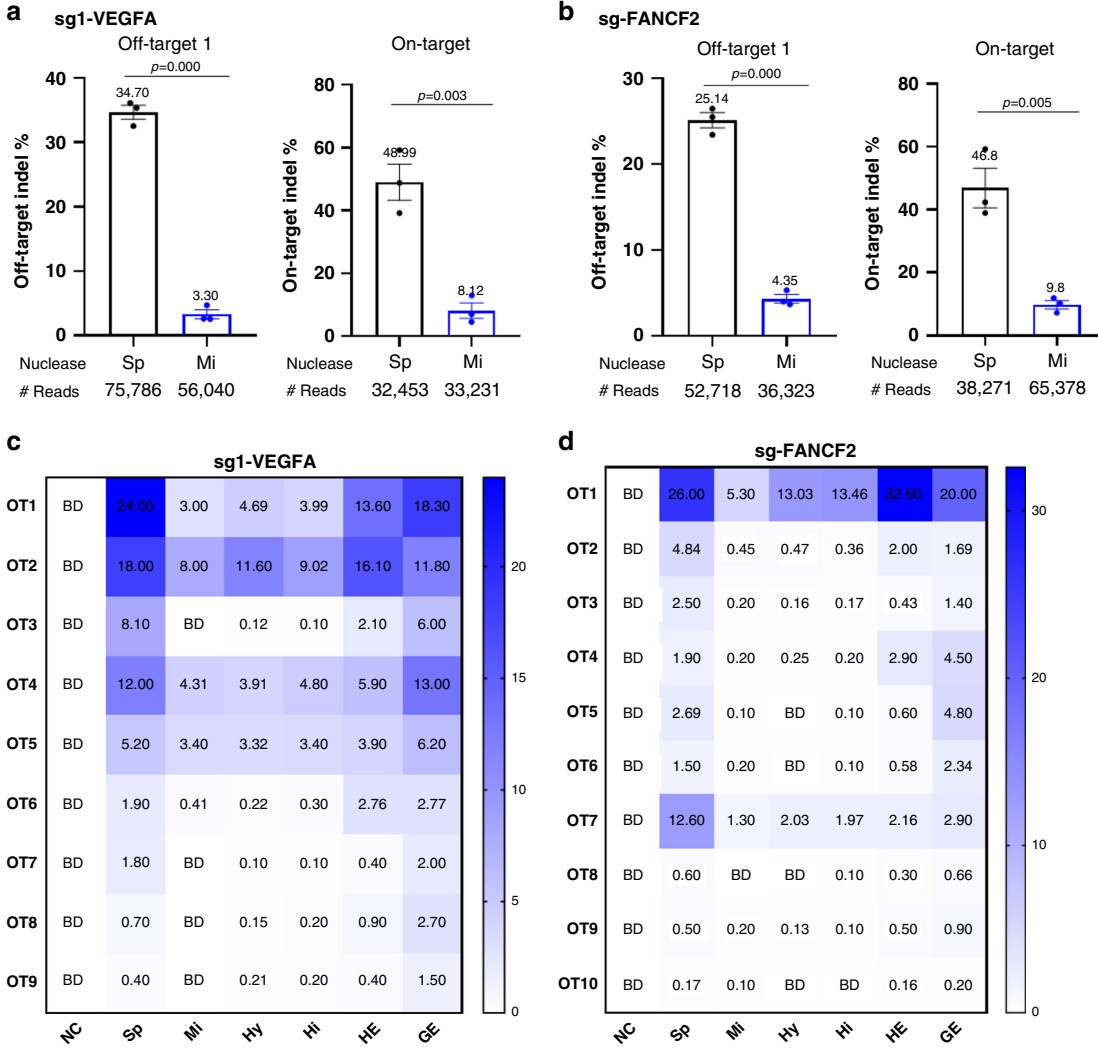

**Fig. 3 MiCas9 reduces off-target indel rates. a** Off-target and on-target indel rates associated with sg1-VEGFA by spCas9 and miCas9 pDNAs. **b** Off-target and on-target indel rates associated with sg-FANCF2 by spCas9 and miCas9 pDNAs. **c** Indel rates at Guide-seq predicted potential off-target loci associated with sg1-VEGFA by pDNAs of different nucleases. **d** Indel rates at Guide-seq predicted potential off-target loci associated with sg-FANCF2 by pDNAs of different nucleases. Sp: spCas9, Mi: miCas9, Hy: hypaCas9, Hi: HiFiCas9, HE: Cas9-HE, GE: Cas9-GE, NC: negative control with non-specific gRNA. #Reads: Average amplicon reads per sample. BD: below detection, representing values <0.10%. Three independent experiments were performed for each condition. Data are presented as mean ± standard error of means (SEM) in **a** and **b**, and as heat map in **c** and **d**. Unpaired *t*-test (two tailed) was used to compare data using GraphPad Prism 8 software (GraphPad Software, Inc., San Diego, CA). Source data are available in the Source Data file.

threshold, representing a 100% reduction. At the other locus SOD1 (1 out of 5, 20%, Supplementary Fig. 7c), however, the on-target indel rates were similar between the spCas9- and miCas9-treated cells (13.47% vs. 14.09%), indicating that miCas9's capacity in reducing on-target indel rates in ss-PGE applications is locus dependent and not universal. Nevertheless, these results show that miCas9 reduces on-target indel at most loci in ss-PGE applications.

We also used miCas9 and spCas9 RNPs, along with gRNA and ssODN donors (Supplementary Table 3) to generate the CCR5 delta32 mutation (CCR5-del32) in hematopoietic stem cells (HSCs), which has been pursued to render HIV-1 resistance in patients[36] (Supplementary Fig. 7d). Consistent with the observations in iPSCs, the PGE rates of CCR5-del32 between these two Cas9 variants are similar (22.0 to 26.5%) in HSCs. Strikingly, the on-target indel rate was reduced to below the detection threshold in the miCas9 group in contrast to 3.2% in the spCas9 group.

These results demonstrate that miCas9 effectively reduces on-target indel rates without compromising efficacy in ss-PGE applications.

**MiCas9 outperforms Cas9-HE and Cas9-GE in ss-PGE applications.** Because miCas9 contains an HDR-enhancing motif Brex27, we next compared it with two representing HDR-fusion Cas9s, Cas9-HE, and Cas9-GE, at four additional PGE loci including HBB, CFTR-G542, CCR5-C101, and PSEN1 using corresponding gRNAs and ss-ODNs (Supplementary Table 3).

Again, miCas9 did not compromise the efficacy of ss-PGE at any of the four loci comparing to spCas9. In fact, at the HBB and CCR5-C101 loci, higher PGE rates were found in the miCas9 group than those in the spCas9 group (Fig. 4c, d and Supplementary Fig. 7e, f), indicating that miCas9 may enhance PGE rates in a locus-dependent manner.

Consistent with earlier findings, the on-target indel rates were significantly reduced by miCas9 at three (HBB, CCR5-C101, and PSEN1) out of the four loci (75%) analyzed, by 63.8%, 79.3%, and 46.9%, respectively (Fig. 4c and Supplementary Fig. 7e, f). But at the other locus CFTR-G542 (25%), the on-target indel rates were comparable between spCas9 (32.6%)- and miCas9 (26.5%)-treated cells (Fig. 4d).

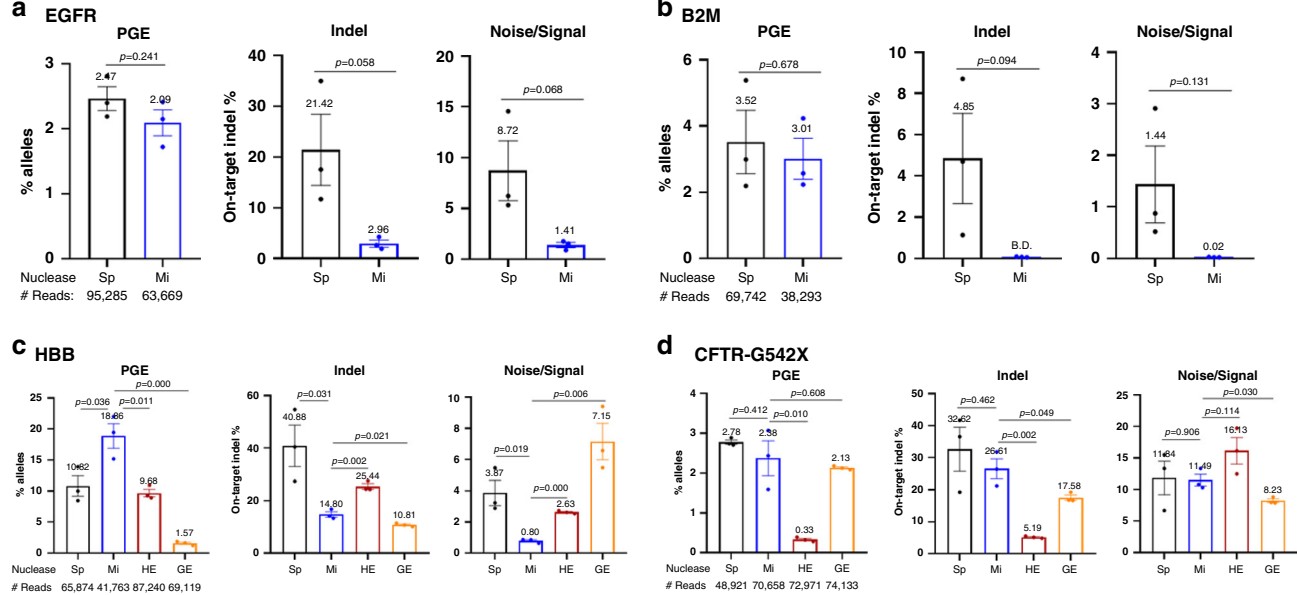

**Fig. 4 Effects of miCas9 on ss-PGE applications. a** PGE rates, on-target indel rates, and the noise/signal ratio at EGFR by spCas9 and miCas9 in human iPSCs. **b** PGE rates, on-target indel rates, and noise/signal ratio at B2M by spCas9 and miCas9 in human iPSCs. **c** PGE rates, on-target indel rates, and noise/signal ratio at HBB by different nucleases in human iPSCs. **d** PGE rates, on-target indel rates, and noise/signal ratio at CFTR-G542 by different nucleases in human iPSCs. Sp: spCas9, Mi: miCas9, HE: Cas9-HE, GE: Cas9-GE. #Reads: Average amplicon reads per sample. Three independent experiments were performed for each condition. Data are presented as mean ± standard error of means (SEM). Unpaired *t*-test (two tailed) was used to compare data using GraphPad Prism 8 software (GraphPad Software, Inc., San Diego, CA). Source data are available in the Source Data file.

Interestingly, the effects of Cas9-HE and Cas9-GE on ss-PGE applications appear different from theirs on ds-KI applications where both GENs enhanced ds-KI rates but had no effects in reducing on-target indels. At these ss-PGE loci, Cas9-HE lowered on-target indel rates consistently; however, at the cost of reduced PGE rates (Fig. 4c, d and Supplementary Fig. 7e, f). Cas9-GE also appeared capable of reducing on-target indel rates, but with an inconsistent PGE outcome. For example, at the PSEN1 locus (Supplementary Fig. 7f), Cas9-GE achieved similar PGE rates, on-target indel rates, and hence an N/S ratio like those by miCas9. However, at the HBB locus (Fig. 4c), Cas9-GE led to significantly lower PGE rates when comparing to all other three (i.e. spCas9, miCas9, and Cas9-HE). These results show that Cas9-HE and Cas9-GE, unlike miCas9, are associated with compromised PGE rates in ss-PGE applications.

Together, these data demonstrate that miCas9 universally maintains and sometimes enhances PGE rates, and is capable to reduce on-target indel rates at most loci in ss-PGE applications.

**Brex27 is a plug and play module that is compatible with other Cas9 variants**. Because the Brex27 consists of only 36 AA, we reasoned that this mini motif can be used as a "plug and play" module when fusing to other variants of GENs without introducing substantial size increase. We tested this strategy in the two examples below.

First, we constructed miHiFiCas9 (Fig. 5a) by fusing Brex27 to HiFiCas9 to see if additional reduction of off-target indel rates can be achieved. Indeed, miHiFiCas9 is shown to reduce the off-target indel rates of sg-FANCF2 (Supplementary Table 2) caused by spCas9 (26.53%), HiFiCas9 (13.13%) or miCas9 (8.18%) to a lower level (3.06%) (Fig. 5b), in which all Cas9 variants were delivered as pDNAs to Ad293 cells. This is an additional 76.7% reduction from that by HiFiCas9 or 62.6% reduction from that by miCas9, indicating a synergistic effect of the miCas9 and HiFiCas9 designs.

In another effort, we constructed miCas9mSA (Fig. 5a) by incorporating Brex27 to spCas9mSA, which was reported to increase large size gene KI rates by enriching Biotin-labeled donor templates at DSB sites[16], to investigate if this nuclease would further increase KI rates. We compared the success rates of knocking in the biotin-labeled ds-KI GFP construct (GFP-donor-1k-ROSA, Supplementary Table 1) to the ROSA26 locus in fibroblast cells using sg-ROSA (Supplementary Table 1), and Cas9s in their pDNA form. Flow cytometry results show that the highest percentage of GFP+ cells is achieved by miCas9mSA (5.50%), followed by spCas9mSA (4.31%), miCas9 (2.73%), and the lowest in spCas9 (1.45%) (Fig. 5c). These results demonstrate the additive beneficial effects of the Brex27 motif to spCas9mSA.

Together these results demonstrate that Brex27 can be used as a plug and play module that is compatible with other Cas9 variants

**The improvement of miCas9 is Brex27 and RAD51 dependent**. Based on the design, miCas9 binds RAD51 through the Brex27 motif. To confirm this, we first constructed a mutant miCas9 (miCas9-mut) where a point mutation is introduced to the Brex27 coding sequence leading to a mutant peptide (Supplementary Fig. 8). As expected miCas9-mut lost the capacity to reduce on-target or off-target indel rates (Fig. 6a). On the other hand, the use of a RAD51 small-molecule inhibitor B02 (ref. [37]) also eliminated the indel reduction capacity of miCas9 (Fig. 6b). These results show that both Brex27 and RAD51 are required for miCas9's beneficial effects.

To prove that miCas9 binds to RAD51 directly, we performed co-immunoprecipitation (co-IP) in Ad293 cells treated with sg-AAVS1, GFP-donor-1k-AAVS1 donor templates (Supplementary Table 1) along with spCas9 or miCas9 pDNAs. Cas9s were pulled down by the anti-Flag antibodies followed by blotting with anti-RAD51 antibody. As shown in Fig. 6c, left panel, there are significantly higher levels of RAD51 signal detected in the miCas9 group than in the spCas9 group. In another set of experiment, the

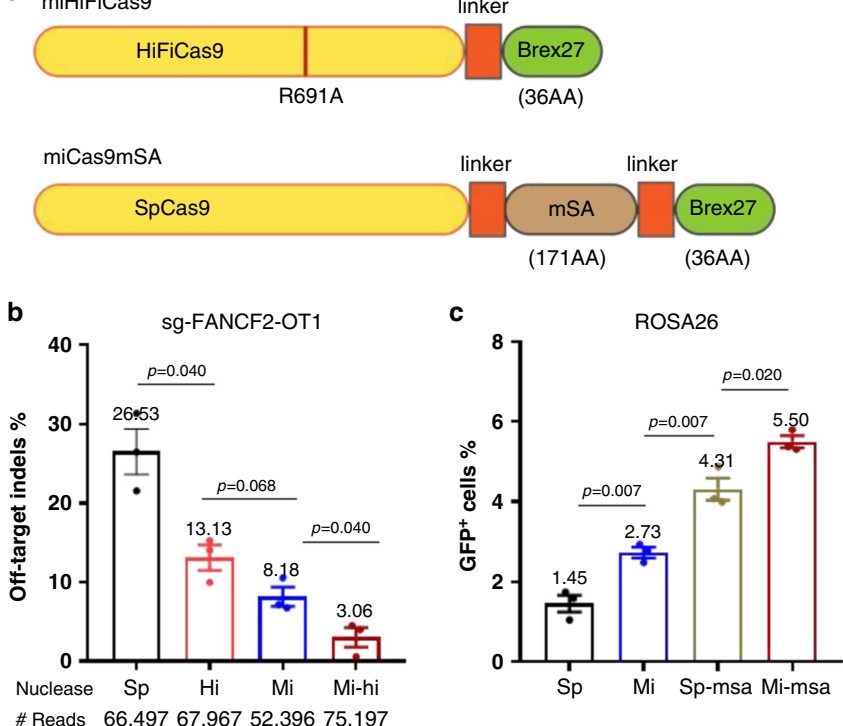

**Fig. 5 Brex27 serves as a plug and play module. a** Illustration of the miHiFiCas9 and miCas9mSA design. **b** Off-target indel rates of sg-FANCF2 by spCas9, HiFiCas9, miCas9, or miHiFiCas9. **c** Percentage of GFP⁺ cells after GFP ds-KI at the ROSA26 locus using spCas9, miCas9, spCas9mSA, or miCas9mSA. Sp: spCas9, Mi: miCas9, Hi: HiFiCas9, Mi-hi: miHiFiCas9, Sp-msa: spCas9mSA, Mi-msa: miCas9mSA. #Reads: Average amplicon reads per sample. Three independent experiments were performed for each condition. Data are presented as mean ± standard error of means (SEM). Unpaired *t*-test (two-tailed) was used to compare data using GraphPad Prism 8 software (GraphPad Software, Inc., San Diego, CA). Source data are available in the Source Data file.

co-IP was performed by anti-RAD51 antibody followed by blotting with anti-Cas9 antibody (Fig. 6c, right panel). Again, high level signals were detected in the miCas9 group where only background signals were detected in the spCas9 group. These results demonstrate that miCas9 directly binds to RAD51.

We then conducted the chromatin immunoprecipitation (ChIP) assay to assess if RAD51 were recruited and enriched by miCas9 at the target locus, also in Ad293 cells treated with sg-AAVS1, GFP-donor-1k-AAVS1 donor templates (Supplementary Table 1). As shown in Fig. 6d, miCas9 significantly enhanced the recruitment of RAD51 to the proximal region of the targeted locus than spCas9 did.

These results confirm that miCas9 relies on Brex27 for binding with RAD51 to achieve the beneficial effects.

## Discussion

In the present work, we show that fusing a minimal motif Brex27 to spCas9 renders the variant miCas9 several desirable features: (i) improving ds-KI rates by multiple folds; (ii) reducing off-target indel events; (iii) maintaining or increasing ss-PGE rates; and (iv) reducing on-target indel rates in both ds-KI and ss-PGE applications. These benefits are consistent with the designing rationale that Brex27 reportedly binds to RAD51 and plays key roles in stabilizing the RAD51/ssDNA nucleoprotein filaments[30], and are corroborated by a recent report in which a Rad51-Cas9 nickase fusion protein mediated HDR while minimizing indels[38].

Adding Brex27 (36 AA) to spCas9 (1368 AA) increases its size by 2.6%. In contrast, other HDR-fusion motifs such as Geminin/GE (110 AA), CtIP/HE (296 AA), mSA (171 AA), DN1S (414 AA), and RAD52 (471 AA) are 306 to 1308% the size of Brex27. Despite its small size, Brex27 has significant HDR-enhancing

capacity, leading to ds-KI rates that are 2–3 folds those by spCas9, similarly to those achieved by Cas9-HE and -GE.

The mini size of Brex27 may prove advantageous especially for in vivo gene editing therapy. Currently, adeno-associated virus (AAV) is the only viral vehicle approved by the U.S. Food and Drug Administration (FDA) for gene therapy[39], which has a packaging capacity of 4.7 kb, which is barely sufficient for packaging spCas9 (4.1 kb). Therefore, any substantial size increase in a Cas9 fusion protein variant will likely affects its eligibility for AAV-mediated delivery, whereas any "size-saving" on the Cas9 part means more packaging space for other elements. In this regard, miCas9 represents a promising variant for AAV-mediated gene editing therapy.

Importantly, at Guide-seq predicted loci for sg1-VEGFA and sg-FANCF2, the mini size Brex27, but not the fusion motifs used in Cas9-HE and Cas9-GE, led to the highest reduction of the off-target indel rates (Fig. 3c, d and Supplementary Figs. 5 and 6) to a level comparable or even lower than those by other specificity-improving Cas9s such as HiFiCas9 and HypaCas9 (Fig. 3c, d). MiCas9 likely shares Guide-seq predicted off-target loci with those by spCas9 because no mutations are made in the Cas9 protein and no sequence or structural changes are made to the sgRNAs. However, there is the remote possibility that Cas9 variants such as miCas9 may induce their own characteristic sets of off-target loci. Follow-up studies may be needed to evaluate this, especially in any clinical-oriented applications.

It should be noted that miCas9 and specificity-improving Cas9s reduce off-target indel events through different mechanisms. MiCas9 works by increasing HDR after DSB generation, whereas specificity-improving Cas9s work by reducing non-specific cuts at the off-target sites. Hence it is conceivable that the

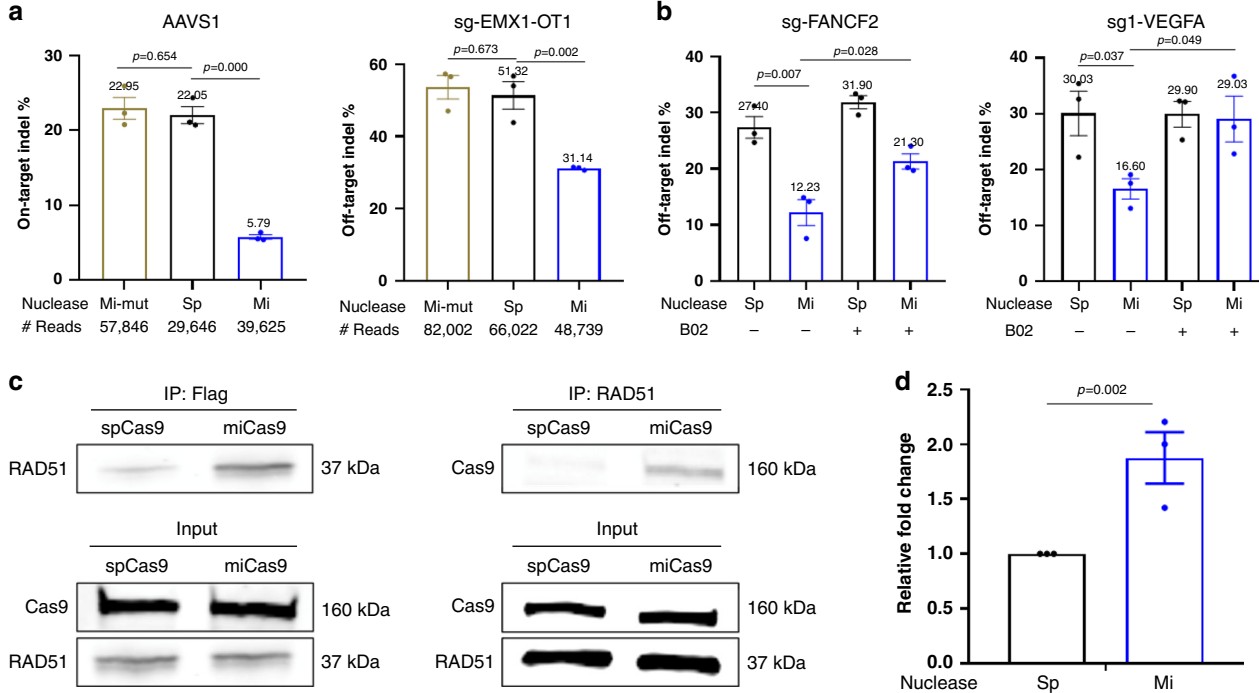

**Fig. 6 MiCas9's beneficial effects are Brex27 and RAD51 dependent. a** Indel rates by spCas9, miCas9, and miCas9-mutant at the on-target site for sg-AAVS1 and the off-target site 1 for sg-EMX1. **b** Indel rates estimated by T7E1 assay by spCas9 and miCas9 with or without the use of RAD51 inhibitor, B02. **c** Co-IP results by using anti-flag (left) or anti-RAD51 (right) antibodies followed western blot for RAD51 or Cas9. **d** ChIP assay results with anti-RAD51 antibody at the proximal region of the sg-AAVS1 target locus. Sp: spCas9, Mi: miCas9, Mi-mut: miCas9-mutant. [#]Reads: Average amplicon reads per sample. Three independent experiments were performed for each condition. Data are presented as mean ± standard error of means (SEM). Unpaired t-test (two tailed) was used to compare data using GraphPad Prism 8 software (GraphPad Software, Inc., San Diego, CA). Source data are available in the Source Data file.

miCas9 strategy is compatible and may be synergistic with that of specificity-improving Cas9s. Indeed, as demonstrated in the present work, miHiFiCas9 was shown to further reduce off-target indel rates up to 77% over HiFiCas9 or up to 63% over miCas9, indicating an apparent synergistic effect by increased specificity and enhanced HDR.

One unique advantage of miCas9 is its capacity to reduce on-target indel rates in both ds-KI and ss-PGE applications, which does not come at the cost of reduced efficacy evidenced by improved ds-KI rates and uncompromised ss-PGE rates in the present study. On-target indel damage is a potential concern of gene editing although many studies have neglected attention to this genotoxic effect, which in clinical settings, should be at the forefront. It is important not to introduce new on-target indel mutations when trying to correct disease-causing mutations, for example on autosomal-dominant mutations or on tumor-associated genes. In comparison with spCas9, miCas9 universally reduces on-target indel rates by 52–75%, in all ds-KI loci ($n = 6$) examined. The reductions are 52–74% and 63–72%, comparing to Cas9-HE and Cas9-GE, respectively.

In ss-PGE applications, miCas9 was effective in reducing on-target indels, as much as 100%, on most of the loci (8 out of 10, 80%) examined, although we did notice exceptions (2 out of 10, 20%) where the on-target indel rates were similar between miCas9 and spCas9. This may indicate that at these two loci (e.g. SOD1 and CFTR-G542) a RAD51-independent mechanism (e.g. SSA) dominates the repair. Nevertheless, to our knowledge, miCas9 represents the smallest and only HDR-improving spCas9 variant to systematically address the on-target damage genotoxicity without comprising efficacy, setting a foundation for follow-up research towards the ultimate goal to abolish/minimize all undesirable on-target indel events.

As demonstrated in the present work, we expect that miCas9 brings benefits to ds-KI and most ss-PGE applications. It is not suitable however if the purpose is to create gene knockout as a result of indel-caused premature stop codon. MiCas9 is expected to work synergistically with gRNA-based improving strategies, for example, the RNA aptamer–streptavidin strategy[40]. The use of Brex27 can go beyond spCas9, as long as the fusion partner works in a RAD51-dependent manner, which would include most DSB-generating GENs, for example, the *Staphylococcus aureus* Cas9 (saCas9)[41], for which extensive engineering and evolution work has yet to be conducted.

In summary, we show that a rationally designed HDR-fusion Cas9 variant miCas9 possess a unique combination of desirable features including improving ds-KI rates, reducing undesirable off-target events, and reducing on-target indel events in ds-KI and ss-PGE applications, providing a "one small stone for three birds" tool in gene editing. The mini size of Brex27 allows it to work as a plug and play module that will benefit many other GEN variants. MiCas9 and the Brex27 module may find broad applications in gene editing research and therapeutics.

## Methods

**Cells**. Human fibroblast cells (Cat# CRL2522) were acquired from American Type Culture Collection (ATCC, Manassas, VA). Cells were cultured with RPMI-1640 Medium (Cat# 11875119; ThermoFisher Scientific, Waltham, MA) supplemented with 10% fetal bovine serum (FBS; Cat# FBS1824-001, Nucleus Biologics, San Diego, CA).

Human AECs were obtained from ATCC (Cat# PCS-300-010) and cultured with Airway Epithelial Cell Basal Medium (Cat# PCS-300-030, ATCC) supplemented with Bronchial Epithelial Cell Growth Kit (Cat# PCS-300-040, ATCC).

Human iPSC cells (Cat# ACS-1030, ATCC) were cultured in feeder-free condition in mTeSR1 (Cat# 85850, StemCell Technologies, Vancouver, Canada) on Matrigel (Cat# 354277, Corning Inc, Corning, NY)-coated cultureware surface.

Human Ad293 cells (Cat# 240085, Agilent, Santa Clara, CA) were cultured in Dulbecco's modified Eagle's medium (DMEM) (Cat# 11995-065, ThermoFisher Scientific) supplemented with 10% fetal bovine serum.

Human CD34[+] HSCs (Cat# PCS-800-012, ATCC) were cultured in Steriflip-filtered StemSpan SFEM II medium (Cat# 09655, StemCell Technologies) supplemented with SCF (100 ng/ml), TPO (100 ng/ml), Flt3L (100 ng/ml), IL-6 (100 ng/ml), UM171 (35 nM), 20 mg/ml streptomycin, 20 U/ml penicillin, and SR1 (StemRegenin1; 0.75 μM).

Human Jurkat cells (Cat# TIB-152, ATCC) were cultured in RPMI-1640 medium (Cat# 30-2001, ATCC) supplemented with 10% FBS.

**Cas9s and guide RNAs.** Unless otherwise specified in the context, "Cas9" in the "Methods" section refers to any Cas9 variant (e.g. spCas9, miCas9, miCas9mSA, miHiFiCas9, HypaCas9) that is used in a given experiment.

To construct miCas9, dead-spCas9-VP64 (Cat# 47754, Addgene) was reverse engineered to spCas9-VP64 by generating the reverse mutations of A10D and A840H, followed by replacement of VP64 with the Brex27 sequence. The spCas9-expressing plasmid DNA was constructed on the same backbone by removing the Brex27 sequence.

To construct spCas9mSA or miCas9mSA, the mSA coding sequence was amplified from pDisplay-mSA-EGFP-TM (Cat# 39863, Addgene) and inserted to the C-terminus of spCas9 or miCas9 with optimized linkers on both sides.

To construct HiFiCas9 and miHiFiCas9 plasmids, we generated the R691A mutation on the spCas9 and the miCas9 plasmids. HypaCas9 (Cat# 101178), Cas9-HE (Cat# 109400), and Cas9-GE (Cat# 109401) plasmid DNAs were acquired from Addgene (www.addgene.org).

To enable RNP experiments using miCas9, we produced recombinant miCas9 protein from E. coli. The miCas9 DNA sequence with 3′ SV40 nuclear localization signal (NLS) was obtained by PCR amplification and was subcloned into pET30 vector in frame with a 5′ His-tag to construct the pET30-miCas9 vector, which was then used to transform BL21(DE3) competent E. coli (Cat# C600003, ThermoFisher Scientific). MiCas9 proteins were His-tag purified, followed by removal of endotoxin by High Capacity Endotoxin Removal Resin (Cat# 88271, ThermoFisher Scientific). Purified miCas9 proteins were stored in 50% glycerol at −20 °C. Cas9-HE and Cas9-GE were similarly in-housed produced.

For synthesis of sgRNA-expressing plasmid, annealed DNA oligonucleotide duplexes were ligated into phU6-sgRNA (Cat# 53188, Addgene). For RNP experiments, sgRNAs were in vitro transcribed by using gRNA synthesis kit (Cat# A29377, ThermoFisher Scientific).

**KI donor templates.** The GFP-expressing ds-DNA donor templates consist of GFP coding sequences flanked by left and right homologous arms corresponding to target sites (Supplementary Table 1). The single-stranded oligonucleotide donors were commercially synthesized by IDT. 5′-Biotin-modified primers were used for PCR to produce biotinylated GFP-expressing donor templates in the miCas9mSA experiments.

**Electroporation.** We used a tube electroporation machine (Model#CTX-1500A LE, Celetrix, Manassas, VA) to deliver Cas9 elements (e.g. Cas9 pDNA or RNP, gRNAs, and donor templates) to cells. $2–3 \times 10^6$ cells were resuspended in 120 μL electroporation buffer (Cat#13–0104, Celetrix). To deliver Cas9 in plasmid DNA form, 4.5 μg Cas9-expressing pDNAs, 1.5 μg sgRNA-expressing pDNAs, and 4 μg GFP-expressing donor templates were added to the buffer. To deliver Cas9 in RNP form, 10 μg Cas9 pre-mixed with 3.3 μg gRNA were added to the buffer. Four micrograms of GFP-expressing donor templates or 10 μg ssODN donor templates were added in KI experiments.

The electroporation conditions were 620 V for 30 ms for fibroblast cells, AECs iPSCs and Ad293 cells, or 635 V for 30 ms for HSCs.

**T7EI assay.** T7EI (T7 endonuclease I) assay was conducted as previously described[42]. Briefly, the purified PCR products were denatured and re-annealed and digested with T7EI (M0302L, New England BioLabs, Ipswich, MA) for 30 min at 37 °C, and then run in a 2% agarose gel. Non-perfectly matched DNA (presumably indel sites) would be recognized and cleaved by T7EI leading to two cleaved bands, whereas the perfectly matched DNA would not be recognized and cleaved by T7EI hence leading to only one band (the wild-type band). The indel rate (%) was quantified using the formula $100 \times (1 - (SQRT(1 - ((A1 + A2)/(A1 + A2 + A3)))))$, where A3 is the intensity of the wild-type band, A1 and A2 are the intensities of the two cleaved bands, respectively, and SQRT is the square root function.

**Deep sequencing (deep-seq) and analysis.** Cells were harvested 48 h after transfection and genomic DNAs were extracted with the Wizard Genomic DNA Purification Kit (Cat#A1120, Promega, Madison, WI). Targeted regions were PCR amplified using high-fidelity PCR master mix (Cat#F532L, ThermoFisher Scientific) with corresponding primers (Supplementary Table 4). The products were gel purified using Qiaquick gel purification kit (Cat#28706, Qiagen, Germantown, MD). For deep sequencing, the purified PCR products were sent to the CCIB DNA Core Facility at Massachusetts General Hospital (Cambridge, MA). Briefly, Illumina compatible adapters with unique barcodes were ligated onto each sample

during library construction. Libraries were pooled in equimolar concentrations for multiplexed sequencing on the Illumina MiSeq platform with $2 \times 150$ run parameters. Upon completion of the NGS run, data were analyzed, demultiplexed and subsequently entered into an automated de novo pipeline. Analysis was completed through the MGH CCIB's de novo assembler UltraCycler v1.0, manually inspected for quality control. A brief description of the algorithm is available at https://dnacore.mgh.harvard.edu/new-cgi-bin/site/pages/crispr_sequencing_pages/crispr_sequencing_algorithm.jsp. PGE and indel rates were determined by the CRISPRResso2 tool[43] (http://crispresso.pinellolab.partners.org/) using.fastq files. For analysis of the PGE and indel rates at CCR5-del32, where the primer and amplicon sequences are highly homologous with those at CCR2, the.seq files were analyzed by Blastn (https://blast.ncbi.nlm.nih.gov/) against the HDR amplicon sequence and the wild-type amplicon sequence.

**Fluorescence-associated cell sorting.** In the GFP gene KI experiments, 3 days post transfection we disassociated cultured cell into single-cell suspension in 300 μL PBS with 2% FBS, and filtered with 70 μm nylon strainer. The cells were subjected to flow cytometry using the MoFlo Astrios EQs Sorter (Model# B52102, Beckman Coulter, Indianapolis, IN) at the University of Michigan Flow Cytometry Core. The data were analyzed using FlowJo (Version 10, Tree Star, Ashland, OR). Cells were gated base on FSC/SSC, doublet discrimination, live/dead, and then by GFP expression (examined in Supplementary Fig. 9). The percentage of GFP-positive cells were used as the indicator of gene KI events.

**Validation of non-precise integrations at junction parts.** Nest PCR was used to amplify the junction parts in KI experiments. For left junction, first round PCR was performed using primer pair of F1/R3, then second round PCR using primer pair of F1/R1. For right junction, first round PCR was performed using primer pair of F3/R2,and then second round PCR using primer pair F2/R2. The second round PCR products were used for deep sequencing to confirm the precise HR. Primer sequences are provided in Supplementary Table 4.

**ChIP assay.** ChIP assay was performed according to the protocol from SimpleChIP Enzymatic Chromatin IP Kit (Magnetic Beads, CST, #9003S). Briefly, Ad293 cells were transfected with plasmid expressing spCas9 or miCas9 with electroporation and the transfected cells were cultured for 48 h. For crosslinking of the co-factors with DNA, cells were incubated with 1% freshly prepared paraformaldehyde at room temperature for 10 min. For RAD51 crosslinking, cells were treated with 2 mM disuccinimidyl glutarate (DSG, ThermoFisher Scientific, #20593) at room temperature for 45 min before 1% paraformaldehyde incubation for 10 min. Then the crosslinking was quenched by 0.1% glycine at room temperature for 5 min. Nuclei were digested with Micrococcal Nuclease at 37 °C for 20 min followed by sonication (Branson Sonifier SLPe, 20 s of 30% amplification, three times). Chromatin was immunoprecipitated with antibodies against RAD51 (Novus Biologicals, NB100-148, 1:50 dilution) or normal mouse IgG (Sigma 12-371, 1:500 dilution) at 4 °C overnight with rotation. ChIP grade Protein G magnetic beads were added to the IP samples and rotated 2 h at 4 °C. DNA was eluted after washing the beads three times with low-salt wash buffer, one time with high-salt wash buffer, and one time with LiCl buffer. After treatment with Proteinase K at 65 °C for 2 h, the DNA was purified and then amplified by real-time PCR with the following primers around the AAVS1 sgRNA target site: 5′- GGAAGGAGGAGGCCTAAGGA-3′ and 5′-ATGTGGCTCTGGTTCTGGGT-3′.

**Co-immunoprecipitation.** Ad293 cells were transfected with spCas9 or miCas9 as described above. After 72 h, lysed with Pierce IP Lysis Buffer (ThermoScientific, #87788). The cell lysate was sonicated for three times, 5 seconds each using a Branson Sonifier SLPe at 35% amplitude and centrifuged at 4 °C, 21,000 g for 10 min. The cell lysate was incubated with anti-FLAG M2 magnetic beads (Sigma, M8823) at 4 °C overnight. The immunocomplexes were then eluted for western blot analysis for RAD51 (Novus Biologicals, NB100-148, 1:1000 dilution or Abclonal, A6268, 1:1000 dilution). For the reverse immunoprecipitation, the cell lysate after sonication was pre-cleaned with Protein A Magnetic Beads (CST, # 73778) at room temperature for 20 min before incubating with primary antibody against RAD51 (Novus Biologicals, NB100-148, 1:100 dilution, or Abclonal, A6268, 1:100 dilution) at 4 °C overnight with rotation. The immunocomplexes were then precipitated with Protein A Magnetic Beads and eluted for western blot analysis for Cas9 (Abclonal, A14997, 1:2000 dilution).

**Sequence information of guide RNAs, ss-ODNs, ds-KI donors, and primers.** Sequence information of guide RNAs, ss-ODNs, ds-KI donors, and primers are provided in Supplementary Tables 1–4.

**Statistics and reproducibility.** Quantitative data are presented as mean ± SEM. Measurements were taken from three distinct samples. Unpaired t-test (two tailed) was used to compare data using GraphPad Prism 8 software (GraphPad Software, Inc., San Diego, CA). Exact P values are labeled in figures. At least two independent experiments were conducted to confirm the results in Co-IP (Fig. 6c), western blot

(Supplementary Fig. 1b), Coomassie blue staining (Supplementary Fig. 1c), and T7E1 (Supplementary Figs. 1d and 4a, b) assays.

**Reporting summary**. Further information on research design is available in the Nature Research Reporting Summary linked to this article.

## Data availability

Raw data for all figures are available upon reasonable request. All targeted amplicon sequencing data have been deposited at the Sequence Read Archive BioProject with accession number PRJNA615686. Any other relevant data are available from the authors upon reasonable request. MiCas9 plasmid DNA is deposited to Addgene (Plasmid ID#: 162758). Source data are provided with this paper.

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

## Acknowledgements

The work is supported by NIH grants GM122181 and HL138139 to J.Z., OD023194 to J. X., and HL068878 and HL129778 to Y.E.C. This work utilized Core Services supported by Center for Advanced Models for Translational Sciences and Therapeutics (CAM-TraST) at the University of Michigan Medical Center.

## Author contributions

J.X., Y.E.C., and J.Z. conceived the idea. J.X., L.M., J.R., J.S., L.W., D.Y., J.Z., and X.X. conducted experiments. J.X., L.M., Y.E.C., and J.Z. analyzed the data and wrote the manuscript.

## Competing interests

J.X., Y.E.C., and X.X. are equity holders of ATGC Inc., a licensee of the miCas9 technology from the University of Michigan. The remaining authors declare no competing interests.
