## [Peer Review File · Nature Communications]

Reviewers' Comments:

Reviewer #1:

Remarks to the Author:

The revised manuscript by Zhang and colleagues has an improved discussion and presentation of data relative to previous submitted versions of this manuscript. The authors' finding that fusion of a 36-aa motif to SpCas9 can alter knock-in, on-target, and off-target editing frequencies will be of broad interest to the readership of Nature Communications, and this study could serve as a foundation for further work in this area. That said, I still have a concern about the authors' off-target claims and use of the term "safety", which I believe can be addressed without further experiments.

Major Comments:

1a) Rigorous off-target analysis includes explicitly presented control and treated data (including number of sequencing reads) for all replicates tested (either in the main text figure or in the SI). The data in Figures 3C and 3D are not a sufficient summary of the off-target analysis. The data used to generate the P values in Supplementary Tables 5 and 6 need to be presented.

1b) Many more sites (and a more rigorous study) would be needed to support the authors' claim on lines 162-163 that "These results show that miCas9 represents one of the most effective Cas9 variants in reducing undesirable off-target indel rates." The authors should remove this sentence prior to publication, especially since it is not the main point of the study.

2) This manuscript does not explicitly address or define "safety". The title should be changed and the last line of the introductions (line 56) should be modified to avoid use of this term.

Reviewer #2:

Remarks to the Author:

Thank you for the efforts on addressing the comments on the revised manuscript. I think with the revised texts and qualifications on the data interpretation, this current version of manuscript will be suitable for publication. This work will provide a compact approach to mitigate Cas9 indel related effects for applications where this is a significant concern.

RESPONSES TO REVIEWER COMMENTS

Reviewer #1 (R1):

Major Comments:

R1.1a: *Rigorous off-target analysis includes explicitly presented control and treated data (including number of sequencing reads) for all replicates tested (either in the main text figure or in the SI). The data in Figures 3C and 3D are not a sufficient summary of the off-target analysis. The data used to generate the P values in Supplementary Tables 5 and 6 need to be presented.*

Response: Thank you for the insightful suggestion. Data are now presented in Supplementary Figure 5 and 6, replacing Supplementary Tables 5 and 6.

R1.1b: *Many more sites (and a more rigorous study) would be needed to support the authors' claim on lines 162-163 that "These results show that miCas9 represents one of the most effective Cas9 variants in reducing undesirable off-target indel rates." The authors should remove this sentence prior to publication, especially since it is not the main point of the study.*

Response: Thank you for your comment. This sentence is changed to “These results show that miCas9 effectively reduces undesirable off-target indel rates”.

R1.2: *This manuscript does not explicitly address or define "safety". The title should be changed and the last line of the introductions (line 56) should be modified to avoid use of this term.*

Response: Thank you. The title is changed to “Rationally designed miCas9 increases large size gene knock-in rates and reduces undesirable on-target and off-target indel edits”. Line 56 is changed to “...with improved efficacy for increasing ds-KI rates, and reduced off-target and on-target indel events”

Reviewer #2 (R2):

R2.1. *Thank you for the efforts on addressing the comments on the revised manuscript. I think with the revised texts and qualifications on the data interpretation, this current version of manuscript will be suitable for publication. This work will provide a compact approach to mitigate Cas9 indel related effects for applications where this is a significant concern.*

Response: Thank you.

Reviewers' Comments:

Reviewer #1:

Remarks to the Author:

The authors have addressed my comments with their modifications to the text and the addition of Supplementary Figures 5 and 6.

Point by point response to reviewers' comments

Reviewer #1 (Remarks to the Author): The authors have addressed my comments with their modifications to the text and the addition of Supplementary Figures 5 and 6.

Response: Thank you.